# Nano-SiO$_2$-Modified Waterborne Acrylic Acid Resin Coating for Wood Wallboard

Lujing Wu [1,†], Meiling Chen [2,*,†], Jing Xu [1,*], Fang Fang [2], Song Li [3] and Wenkai Zhu [3,*]

1 Academy of Art, Jinling Institute of Technology, Nanjing 211199, China
2 Jiangsu Co-Innovation Center of Efficient Processing and Utilization of Forest Resources, College of Materials Science and Engineering, Nanjing Forestry University, Nanjing 210037, China
3 College of Chemistry and Materials Engineering, Zhejiang A&F University, Hangzhou 311300, China
* Correspondence: meiling_chen@njfu.edu.cn (M.C.); xjj@jit.edu.cn (J.X.); wenkai0814@zafu.edu.cn (W.Z.)
† These authors contributed equally to this work.

**Abstract:** As one of the key products of the whole wood custom home, the study of environmental protection coating technology for wood wallboard has practical significance. Waterborne acrylic acid (WAA), as an important research object of environmentally friendly coatings, has been studied mainly in the area of modification. However, there is less research on its application to the field of wood wallboard. Herein, we developed a facile strategy to prepare WAA resin coatings with excellent performance using SiO$_2$ nanoparticles as modifiers and explored the feasibility of wood wallboard. In this work, a simple mechanical compounding method was used to modify the WAA resin coatings by adding nano-SiO$_2$, aiming to improve their physicochemical properties such as hardness and abrasion resistance while maintaining gloss. It was found that nano-SiO$_2$ showed different effects on the viscosity, gloss, adhesion, and abrasion resistance of WAA resin coating. The combined performance results showed that the wood wallboard finishes exhibit excellent performance when the modifier nano-SiO$_2$ was added at 1 wt% in the WAA resin coating. This present work shows that the nano-SiO$_2$-modified WAA resin coating for wood wallboard has a broad application prospect.

**Keywords:** wooden wallboard; waterborne acrylic acid resin coating; nano-SiO$_2$; coating film properties





## 1. Introduction

Nowadays, a whole wooden household has become a general trend in the industry because it caters to consumers' pursuit of high quality, environmental protection, and a personalized life [1–3]. The whole wooden household can save consumers a lot of trouble when decorating, and the "one-stop" service concept caters to the needs of current consumers. Therefore, many wood companies integrate various resources to launch whole wooden household products to meet consumer demand [4–7]. Moreover, wooden wallboard is exactly one kind of whole wood home improvement product. Finishing the surface of wooden wallboard can not only make up for the defects of natural wood but also give it certain physical and chemical properties. However, the research on the coating of wooden wallboard is still based on traditional coating technology, and the research on the development of environmental protection coatings is still less [8–10]. Besides, the development of water-based resin coatings suitable for wood wallboard needs urgent research [11–15].

With the rising awareness of environmental protection and the strict requirements for environmental quality, the development of water-based environmental protection coatings has been promoted [16–18]. Compared with traditional wood coatings using oil-based paint, water-based environmentally friendly coatings use water as a dispersant or solvent, which enters the air as water vapor in the evaporation process, greatly reducing the emission of volatile organic compounds (VOC) [19,20]. Therefore, the water-based coating ensures that it is environmentally friendly from the source. Furthermore, only resin and some additives of water-based coatings are extracted from petroleum, which saves a lot of petroleum

resources and reduces the dependence on fossil resources [21–23]. In addition, compared with other water-based coatings, waterborne acrylic acid (WAA) resin coating has a lower economy and a certain potential to be applied to the painting of wood wallboard [24–27].

WAA resin coating has performance defects such as weak abrasion resistance, low chemical resistance, and general hardness, while nano-$SiO_2$ has excellent hydrophilicity, stability, and reinforcement [28–30]. As far as we know, the modification of WAA resin coatings by nano-$SiO_2$ can effectively improve the relevant physicochemical properties (such as hardness, gloss, adhesion, and resistance to adhesion) of the coatings [31–35]. Therefore, it is of practical significance to study the coating technology of modified WAA resin coatings by applying them to the painting of wood wallboard.

In this study, we modified the WAA resin coatings with nano-$SiO_2$ using a simple mechanical co-blending method. Subsequently, the modified WAA resin coatings were applied to the finishing of the wood wallboard, and the surface properties of the paint films were investigated. Furthermore, this study also provides a basis and reference for the coating technology of WAA resin as a coating raw material for wood wallboard.

## 2. Materials and Methods

### 2.1. Materials

Silicon dioxide nanoparticles (Nano-$SiO_2$, 30 nm) were supplied by Nanjing Lisheng Chemical Company (Nanjing, China). Waterborne acrylic acid (WAA) was obtained from Changzhou Guangshu Chemical Technology Co., Ltd. (Changzhou, China). A light white oak wall panel (100 × 100 × 18 mm), made of 30 μm thickness of the red oak veneer, was supplied by Dehua TB New Decoration Material Co., Ltd., Huzhou, China. The moisture content of the panel was 11.2%. Distilled water was used in the experiments, and other reagents were used directly without any treatment.

### 2.2. Preparation of Nano-$SiO_2$-Modified WAA Resin Coating

WAA resin coating was reinforced with different content of nano-$SiO_2$ by mechanical blending method. Briefly, weighed 100 g of WAA resin coating into a baker (Jiangsu Lele Teaching Equipment Co., Ltd., Taizhou, China) and added 15 g of deionized water for dilution. Subsequently, the nano-$SiO_2$ was added to the resin emulsion at 0.5 wt%, 1 wt%, 1.5 wt%, 2 wt%, 2.5 wt%, and 3 wt% of the WAA mass. Then, stirred with a magnetic stirrer (RH BASIC S025, Aika Instrument Equipment Co., Ltd., Guangzhou, China) at 500 r/min for 2 h at room temperature to make the dispersion uniform (the specific formulation is shown in Table 1). Finally, it was placed in an ice bath and sonicated with an ultrasonic crusher (NY-JY98-IIIDN, Changzhou Empei Instrument Manufacturing Co. LTD, Changzhou, China) for about 2 min to make the nano-$SiO_2$ uniformly dispersed in the resin emulsion.

**Table 1.** The proportion of nano-$SiO_2$ in WAA resin in different samples.

| Component ╲ Samples | 1 | 2 | 3 | 4 | 5 | 6 |
|---|---|---|---|---|---|---|
| WAA (g) | 100 | 100 | 100 | 100 | 100 | 100 |
| Deionized water (g) | 15 | 15 | 15 | 15 | 15 | 15 |
| Nano-$SiO_2$ (g) | 0.5 | 1 | 1.5 | 2 | 2.5 | 3 |

### 2.3. Preparation of WAA Finished Wood Wallboard

In order to test the paint film performance of the WAA resin coating modified by nano-$SiO_2$, it was applied on the surface of the wooden wallboard by spraying with a spray gun. Typically, pour the modified waterborne acrylic topcoat into the spray can of the gun (W-71, IWATA, Fukushima-Ken, Japan) and keep the nozzle discharge to a minimum. Then, spraying wooden wallboard with 4 guns by the "crossed" method (first horizontal spray and then vertical spray), controlling the paint output of 48 g/cm$^2$ per gun (error range is

30–80 g/cm$^2$). Finally, the sprayed specimen was placed in the humidity chamber (LHS-80HC-I, Shanghai Yiheng Scientific Instrument Co., Ltd., Shanghai, China) with forceps for drying (humidity chamber temperature was 25 ($\pm$0.5) °C, relative humidity was 70 ($\pm$1)%), it will be taken out to test the relevant performance after 7 days of drying.

### 2.4. Characterizations

Fourier transform infrared spectroscopy (FT-IR, Cary630, Agilent Technologies Inc., Santa Clara, CA, USA) was carried out to determine the characteristic absorption peak of WAA coating before and after nano-SiO$_2$ modification. The samples were mixed with KBr at a mass ratio of 1:8 in a mortar. After that, the mixture was ground uniformly and pressed under a pressure of 16 MPa. The spectrum was recorded from the wavenumber from 4000 cm$^{-1}$ to 500 cm$^{-1}$ at 2 cm$^{-1}$ resolution with 48 scans. A scanning electron microscope (SEM) was operated to investigate the morphology and structure with a JEOL-7800F (JEOL Ltd., Tokyo, Japan) at an accelerating voltage of 20 kV. The physical and mechanical properties of WAA coating were analyzed in accordance with corresponding Chinese Standards. The gloss of the coating film of WAA before and after nano-SiO$_2$ modification was tested by the glossmeter (YG60S 60°, Voda Precision Instrument Co., Ltd., Wuhan, China) according to GB/T9754-2007. The specific test method is as follows: place the calibrated glossmeter on the coating surface and test 6 values at different parts; calibrate the glossmeter every three data readings. If the difference between the maximum value and the minimum value is less than 20% of the average value, record the data and the average value, otherwise re-select the test piece for testing. An abrasion tester (BGD-523, Biaogeda Precision Instrument Guangzhou Co., Ltd., Guangzhou, China) was carried out to investigate the abrasion resistance of wood wallboard coated with WAA coating reinforced by nano-SiO$_2$. We evaluated the abrasion resistance of painted wood wallboard after 100 times of grinding, and the unit of abrasion resistance was mg/100 r according to GB/T1768-2007. Moreover, take the average of three values as the abrasion resistance. The pencil hardness method was used to test the hardness of WAA paint film before and after modification (GB/T23999-2006). A pencil hardness tester (BGD 507/S, Biaogeda Precision Instrument Guangzhou Co., Ltd., Guangzhou, China) with a pencil of a certain hardness was used to scratch 7 mm on the surface of the coating at a speed of 1 mm/s. Meanwhile, the hardness of the pencil was changed from soft to hard, and the hardness of the pencil was indicated when the coating showed 3 mm or more scratches. The adhesion was measured by a paint film scriber (BEVS 2202, BEVS Industrial Co., Ltd., Zhuhai, China) at 2 mm intervals according to GB/T9286-1998, and the adhesion level was evaluated according to the peeling of the paint film.

## 3. Results and Discussion

### 3.1. FT-IR of WAA Resin Coating Modified by Nano-SiO$_2$

The characteristic functional group and characteristic absorption peak of WAA resin before and after nano-SiO$_2$ modification were investigated using FT-IR. The FT-IR spectra of WAA modified with 1.5% nano-SiO$_2$ were used for comparison. It can be seen from Figure 1 that the FT-IR spectra of the WAA coatings showed similar absorption peaks after the modification of nano-SiO$_2$, which indicates that the modification of nano-SiO$_2$ did not destroy the basic structure of the WAA coatings by chemical reaction. The characteristic absorption peaks of the samples were observed at approximate peaks of 3400–3300, 2930, 2850, 1730, 1450, 1400, 1240, and 1140 cm$^{-1}$ (Figure 1). As can be seen from the FT-IR spectrum, the stretching vibration peak near 3400–3300 cm$^{-1}$ was caused by the hydroxyl group in the WAA coating [36]. In addition, additional absorption peaks were observed between 2930 and 2850 cm$^{-1}$ in the FT-IR spectra of the WAA, which corresponded to the antisymmetric stretching vibrational peaks and symmetric stretching vibrational peaks of -CH$_2$ and -CH, respectively [37]. Furthermore, the peaks at 1730, 1450, and 1400 cm$^{-1}$ were attributed to the absorption peaks of C=O, C-O, and -COO- in the carboxyl group of the WAA coating, respectively [38]. Moreover, sharp absorption peaks were shown between

1240 and 1140 cm$^{-1}$, which were ascribed to the antisymmetric stretching vibration peaks and symmetric stretching vibration peaks of ester groups in WAA resin. When SiO$_2$ nanoparticles were added to the WAA resin coatings, no new FT-IR characteristic peaks were seen in the nano-SiO$_2$/WAA coatings, indicating that no chemical reaction occurred to create the new functional group.

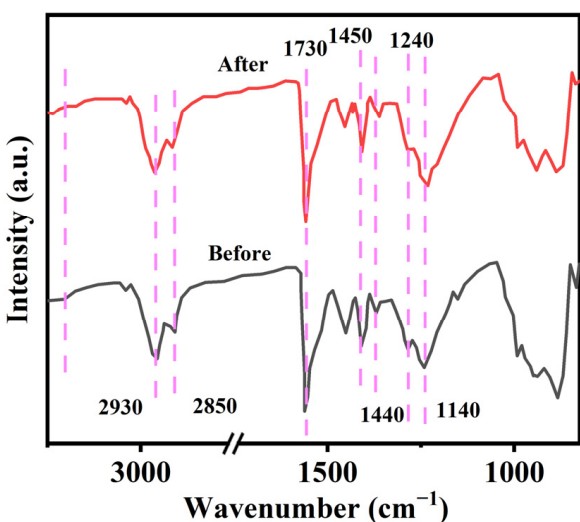

**Figure 1.** The FT-IR spectra of WAA coatings before and after modification by nano-SiO$_2$.

### 3.2. Micromorphological Analysis

Figure 2 shows SEM morphology images of WAA resin before (a and b) and after (c and d) being modified by nano-SiO$_2$. It can be seen from Figure 2a,b that many small particles are dispersed on the surface of the unmodified WAA paint film, which is caused by the pigment particles and other additives in the WAA paint [39]. The irregular lamellar structure can be seen from the cross-sectional view of the pure WAA paint film. Moreover, there are many holes between the lamellae, which are caused by the disordered arrangement of particles in the paint and the air bubbles generated during coating. The existence of these holes makes the lamella loose, which manifests as low hardness and poor wear resistance of the paint film at the macroscopic level.

After the addition of nano-SiO$_2$ to the WAA resin coating, the paint film (Figure 2c,d) was dispersed with small particles similar to the pure WAA paint film surface. However, there are some large particles were also unevenly dispersed on the surface of the WAA paint film after being modified by nano-SiO$_2$. This was attributed to the fact that nano-SiO$_2$ is very easy to agglomerate, and the large particles formed by agglomeration were dispersed in the coating. As can be seen from the modified WAA paint film, there are some large holes and raised large particles distributed at the interface of the paint film (circles in the figure), which are caused by the agglomerated large particles of nano-SiO$_2$ being pulled out when the paint film is sheared. This indicates that nano-SiO$_2$ can be embedded and combined with WAA coatings, and the agglomeration of nano-SiO$_2$ makes its dispersion in WAA coatings poor, which directly affects the appearance and performance of the modified WAA coatings.

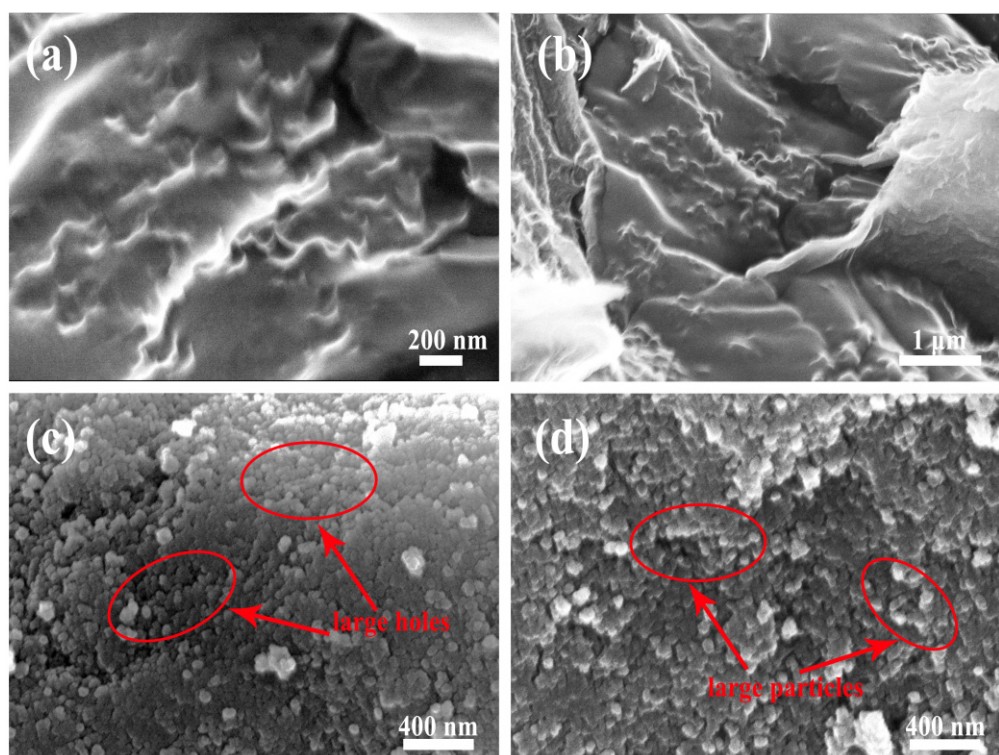

**Figure 2.** The SEM images of WAA films before (**a**,**b**) and after (**c**,**d**) nano-SiO$_2$ modification.

### 3.3. Effect of Nano-SiO$_2$ Addition on the Gloss of WAA Paint Film

The specific effect of nano-SiO$_2$ addition on the gloss of WAA paint film is shown in Figure 3. It can be found from the figure that the gloss of the paint film decreases with the increase in nano-SiO$_2$ addition. The gloss of the paint film was 24.1, 23.1, 20.9, 19.0, 18.7, and 16.8 when the content of nano-SiO$_2$ was increased to 3 wt%, respectively. According to the actual product requirements for the gloss of the paint film for matte, to assess whether it meets the performance of the paint film, 0.5 wt% ≤ nano-SiO$_2$ content ≤ 1.5 wt% is in line with the product requirements.

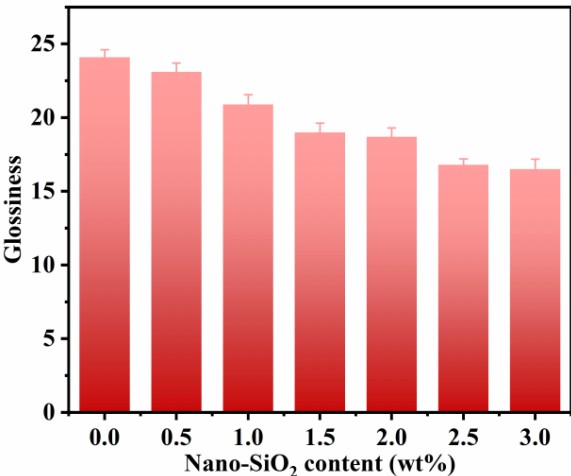

**Figure 3.** Effect of nano-SiO$_2$ content on the glossiness of WAA paint film.

### 3.4. Effect of Nano-SiO₂ Addition on the Abrasion Resistance of WAA Paint Film

The effect of nano-SiO$_2$ addition on the abrasion resistance of WAA coating film is shown in Figure 4. The abrasion resistance of the paint film was evaluated by the weight loss value of 100 r paint film [40]. It can be seen from the figure that the weight loss value of the paint film is negatively correlated with the wear resistance of the paint film when the relevant process conditions are consistent. The 100 r abrasion weight loss value of the paint film was 0.237 g, 0.228 g, 0.214 g, 0.209 g, 0.127 g, 0.117 g, and 0.162 g when the content of nano-SiO$_2$ was increased to 3 wt%, respectively. When the nano-SiO$_2$ addition was 0–2.5 wt%, the basic trend of the wear loss value is reduced, and the best abrasion resistance performance was achieved when the addition amount was 2.5 wt%. However, the weight loss value of the paint film at 0.5 wt% was slightly higher than the wear value when it was not added. The main reason may be that one of the measured specimens had a large abrasion value, which can be largely ignored as abnormal data. When the addition amount reached 3 wt%, the paint film wear weight loss value increased, but it was lower than the pure WAA paint film. This is due to the large specific surface area of nano-SiO$_2$, which caused uneven distribution in the solution and self-agglomeration, resulting in an increase in the abrasion value of the paint film. The addition of nano-SiO$_2$ can improve the hardness of WAA paint film, but when the addition of nano-SiO$_2$ reaches a certain value, the hardness of the paint film is enhanced, the film becomes brittle, and the wear resistance is reduced. Therefore, 1 wt% $\leq$ nano-SiO$_2$ content $\leq$ 2.5 wt% is more in line with the abrasion resistance of WAA paint film.

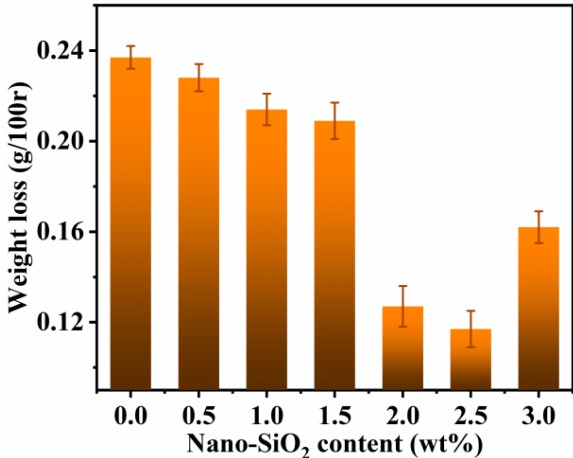

**Figure 4.** Effect of nano-SiO$_2$ content on abrasion resistance of WAA paint film.

### 3.5. Effect of Nano-SiO₂ Addition on the Hardness of WAA Paint Film

The effect of nano-SiO$_2$ addition on the hardness of WAA paint film is shown in Figure 5. The hardness of pure WAA coating can only reach 2B, which does not meet the requirements of the corresponding standard for water-based paint hardness greater than or equal to B. It is obvious that the addition of nano-SiO$_2$ can effectively improve the hardness of paint film. As we know, nano-SiO$_2$ is a nanomaterial with a crystal effect, so adding it to WAA resin can improve the hardness of the coating. When the added amount of nano-SiO$_2$ was in the range of 0.5 wt%–2.5 wt%, the hardness of the paint film was increased to B. When the addition amount is 3 wt%, the hardness of the paint film reaches its best HB. However, according to Figure 4, the corresponding abrasion resistance was the worst at this time. Therefore, a comprehensive assessment of hardness and abrasion resistance with 0.5 wt% $\leq$ nano-SiO$_2$ content $\leq$ 2.5 wt% is more consistent.

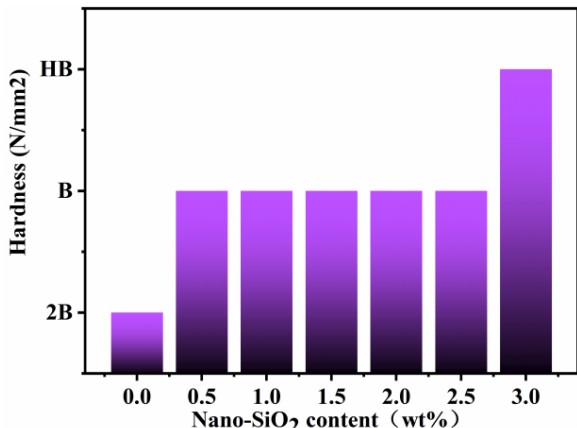

**Figure 5.** Effect of nano-SiO$_2$ content on the hardness of WAA paint film.

### 3.6. Effect of Nano-SiO$_2$ Addition on the Adhesion of WAA Paint Film

The effect of nano-SiO$_2$ addition on the adhesion of WAA paint film is shown in Figure 6. Generally speaking, the adhesion requirements for water-based finishes for wood wallboard are as follows: adhesion (scribe spacing 2 mm/level) less than or equal to 1 level. When the addition amount of nano-SiO$_2$ increased from 0 to 3 wt%, the adhesion grades were 1, 1, 1, 1, 2, 2, and 3, respectively. The adhesion of the coating decreased with the increase in the nano-SiO$_2$ addition, and the adhesion was the worst when the addition amount reached 3 wt%. This was due to there being too many SiO$_2$ nanoparticles that could not be effectively dissolved and leading to structural instability. Although the adhesion grade was grade 1 when the nano-SiO$_2$ content was less than 1.5 wt%. However, the adhesion was the best when no modifier was added, and the adhesion was the worst when the amount of modifier was more than 1.5 wt%. Therefore, 0.5 wt% ≤ nano-SiO$_2$ content ≤ 1.5 wt% is more in line with the adhesion of WAA paint film.

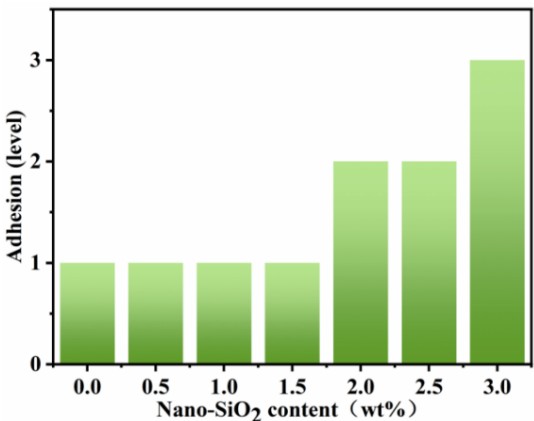

**Figure 6.** Effect of nano-SiO$_2$ content on adhesion of WAA paint film.

### 3.7. Optimal Process

After the above analysis, the amount of modifier nano-SiO$_2$ added to meet the indexes of gloss, hardness, abrasion resistance, adhesion, and cost of the modifier of WAA paint film is shown in Table 2. Taking the above performance factors into account, the film performance of WAA is superior when the content of nano-SiO$_2$ is 1 wt% or 1.5 wt%. Moreover, according to the construction and cost, the construction simplicity when the addition amount is 1 wt% is better than that of 1.5 wt%. This is because the nano-SiO$_2$ in the WAA coating is easily adsorbed to each other and agglomerates. Therefore, with the increase in the amount of nano-SiO$_2$ added, the particles in the WAA coating increase, and the construction difficulty increases. In addition, the cost of adding 1.5 wt% was higher

than that of 1 wt%. Therefore, combining the above factors, the modified WAA coating prepared at 1 wt% nano-$SiO_2$ addition has excellent performance.

**Table 2.** The optimal amount of nano-$SiO_2$ addition for each performance index.

| Performance Index | Nano-SiO$_2$ Addition Conforming to the Process |
|---|---|
| Glossiness | 0.5 wt%, 1 wt%, 1.5 wt% |
| Hardness | 0.5 wt%, 1 wt%, 1.5 wt%, 2 wt%, 2.5 wt%, 3 wt% |
| Abrasion resistance | 1 wt%, 1.5 wt%, 2 wt%, 2.5 wt% |
| Adhesion | 0.5 wt%, 1 wt%, 1.5 wt% |
| Cost | 0.5 wt% < 1 wt% < 1.5 wt% < 2 wt% < 2.5 wt% < 3 wt% |
| Optimal process | 1 wt% |

### 4. Conclusions

In this study, a simple mechanical compounding method was used to modify the WAA coating with nano-$SiO_2$ serving as a modifier. Meanwhile, the feasibility of using it on the surface of wood wallboard to enhance its paint film properties was investigated. The FT-IR spectrum results showed that the modification of WAA coatings with nano-$SiO_2$ did not produce new chemical functional groups, indicating that the basic structure of WAA resin was not destroyed. Thus, the basic properties of nano-$SiO_2$ were maintained. In addition, it is clear from the SEM images that the addition of nano-$SiO_2$ changes the dispersion properties of WAA coatings, which, in turn, affects the performance of the corresponding coatings. Moreover, the hardness, gloss, adhesion, and wear resistance of WAA after being modified by nano-$SiO_2$ showed different degrees of enhancement. This indicates that the WAA coating modified by nano-$SiO_2$ is beneficial for enhancing the paint film properties on the wood wallboard surface. Based on various convenient factors, the properties of the paint film coating on the wooden wallboard can reach their best performance when the content of nano-$SiO_2$ in the WAA coating is 1 wt%. This work demonstrates the potential application of nano-$SiO_2$ in WAA coatings and provides a new perspective for the design of new formaldehyde-free waterborne coatings.

**Author Contributions:** Conceptualization, J.X. and W.Z.; methodology, L.W., F.F. and S.L.; software, M.C. and S.L.; validation, L.W., M.C., J.X. and W.Z.; formal analysis, L.W. and M.C.; investigation, L.W.; resources, M.C.; data curation, S.L.; writing—original draft preparation, L.W. and M.C.; writing—review and editing, J.X. and W.Z.; visualization, M.C.; supervision, J.X.; project administration, J.X.; funding acquisition, J.X. All authors have read and agreed to the published version of the manuscript.

**Funding:** This research was funded by the Research and Development Fund of Zhejiang A&F University (2022LFR076), the Natural Science Foundation of the Higher Education Institutions of Jiangsu Province (19KJB220006), the 2021 project of the "fourteenth five year plan" of Education Science in Jiangsu Province "reform and innovation of professional art talent training mode" (T-c/2021/106), the 2019 Jiangsu University Philosophy and social sciences research project "analysis of typical characteristics of Republic of China home design and Its Enlightenment to contemporary" Republic of China style "design" (2019sja0482), and the achievements in the construction of national first-class undergraduate major in environmental design.

**Institutional Review Board Statement:** Not applicable.

**Informed Consent Statement:** Not applicable.

**Data Availability Statement:** The data that support the findings of this study are available from the corresponding author, J.X., upon reasonable request.

**Acknowledgments:** We would like to thank other members in our groups for helping us prepare for the samples. Advanced analysis and testing center of Zhejiang A&F University is acknowledged.

**Conflicts of Interest:** The authors declare no conflict of interest.

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
