# Peer review of "Nano-SiO2-Modified Waterborne Acrylic Acid Resin Coating for Wood Wallboard"

_coatings, doi:10.3390/coatings12101453_

Round 1

Reviewer 1 Report

Why have the authors used exactly Si02 to improve which property in the research design as it is not clear from the Abstract and intro?

Si02 as glass type of coating can bring different properties but these must the thoroughly  studied instead of lets mix some Si02 into waterborne coating approach. 

For all measurement methods please describe appropriate international or national standards (gloss for example);

For all measurements how many repeats were made? The are std bars but no info on sample number. 

For weight loss the trend is clearly faulty (measurement were needed to be repeated) as preparation mistake is most probable.

FTIR was useless as no interactions between Si02 and waterborne coating is expected.

In conclusions formaldehyde free coating are mentioned, head to head comparison is needed regarding all properties to conclude this kind of claims. Basically a reference sample, or mixing several different waterborne coatings would much improve this paper.

Author Response

Response to Reviewer 1 Comments

On behalf of my co-authors, we thank you very much for giving us an opportunity to revise our manuscript. We appreciate you very much for your positive and constructive comments and suggestions on our manuscript entitled “Nano-SiO2 Modified Waterborne Acrylic Acid Coating for Wood Wallboard”. (ID: coatings-1917973).

We have studied your comments carefully and have made revisions which are marked in red on the paper. Those comments are all valuable and very helpful for revising and improving our paper, as well as the important guiding significance to our research. The main corrections in the paper and the responses to the reviewer’s comments are as flowing:

Point 1: Why have the authors used exactly SiO2 to improve which property in the research design as it is not clear from the Abstract and intro?

Response 1: Thank you for your valuable suggestions. And we have added the use of nano-SiO2 to improve the specific properties of waterborne acrylic acid coatings in the Abstract and Introduction.

Point 2: SiO2 as glass type of coating can bring different properties but these must the thoroughly studied instead of lets mix some SiO2 into waterborne coating approach.

Response 2: We are grateful for the suggestion. It’s true that nano-SiO2 can bring different properties. However, related research have added nano-SiO2 to coatings and enhanced related properties. Therefore, it is of practical significance for this work to mix nano-SiO2 into water-based acrylic coatings for its modification.

Point 3: For all measurement methods please describe appropriate international or national standards (gloss for example);

Response 3: As the Reviewer suggested, we have described the corresponding national standards for the measurement methods in the article.

Point 4: For all measurements how many repeats were made? The are std bars but no info on sample number.

Response 4: Thank you for your comment. We have supplemented the number of replicates for all measurements in the article. Meanwhile, we have added information on sample numbers in Section 2. Materials and Methods.

Point 5: For weight loss the trend is clearly faulty (measurement were needed to be repeated) as preparation mistake is most probable.

Response 5: We are very sorry for our negligence regarding the data on weight loss, and we have revised Figure 4.

Point 6: FTIR was useless as no interactions between SiO2 and waterborne coating are expected.

Response 6: As for the referee’s concern, we believe that FTIR test characterization is necessary, which theoretically proves that there is no interaction between nano-SiO2 and waterborne coatings instead of “expected”.

Point 7: In conclusion, the formaldehyde-free coating is mentioned, head to head comparison is needed regarding all properties to conclude this kind of claim. Basically, a reference sample or mixing of several different waterborne coatings would improve this paper.

Response 7: Thank you for your valuable suggestions, which will guide us in the right direction for our next research. In our future papers, we will study nano-SiO2 mixed with several different waterborne coatings and investigate the related performance of its paint film. But this paper focuses on the feasibility of nano-SiO2 reinforced waterborne coatings. However, in future research, we will focus on the possibility of nano-SiO2 to reinforce other waterborne coatings. We hope you can continue to pay attention to our future research.

Reviewer 2 Report

The concept of this research by focusing on waterborne coating with properties enhancer sounds good but lack of evidence to support the claim as follows:

1.     In materials part, nano-SiO2 with agglomerate size about 30 micron was employed. How did authors ensure about the disaggregation technique used? To prove nanosized particle the several techniques including TEM, nanosizer and size distribution, stability (sedimentation) are necessary.

2.     WAA/nano SiO2 nanocomposite must be achieved in order to obtain testing properties (abrasion resistance, gloss, hardness, chemical resistance, impact resistance etc). But authors failed to achieve nanocomposite preparation judged from FTIR. This indicated that SiO2 still existed in micron sized range and likely to pricipate upon standing (sedimentation test)

3.     After drying (curing) polysiloxane network is expected. No evidence provided (TEM of coating film)

4.     Many more..

So I feel that results are not enough to be published at this time being.

Author Response

Response to Reviewer 2 Comments

Thank you for your comments concerning our manuscript entitled “Nano-SiO2 Modified Waterborne Acrylic Acid Coating for Wood Wallboard” (ID:coatings-1917973). Those comments are all valuable and very helpful for revising and improving our paper, as well as the important guiding significance to our research. We have studied the comments carefully and have made corrections which we hope meet with approval. Revised portions are marked in red on the paper. The main corrections in the paper and the responses to the reviewer’s comments are as flowing:

Point 1: In materials part, nano-SiO2 with agglomerate size about 30 micron was employed. How did authors ensure about the disaggregation technique used? To prove nanosized particle the several techniques including TEM, nanosizer and size distribution, stability (sedimentation) are necessary.

Response 1: On behalf of my co-authors, we thank you very much for giving us an opportunity to revise our manuscript. Your comment is valuable and very professional for revising and improving the manuscript. We have studied this comment carefully and have revised the relevant part about the characterization of nano-SiO2. Moreover, the nano-SiO2 used in our experiment was purchased from the market, so we do not think it is necessary to do further characterization. We can find out about it from the manufacturer.

Point 2: WAA/nano SiO2 nanocomposite must be achieved in order to obtain testing properties (abrasion resistance, gloss, hardness, chemical resistance, impact resistance etc). But authors failed to achieve nanocomposite preparation judged from FTIR. This indicated that SiO2 still existed in micron sized range and likely to pricipate upon standing (sedimentation test)

Response 2: Thank you for your valuable suggestions. However, we believe that in order to obtain the excellent physical properties (abrasion resistance, gloss, hardness, chemical resistance, impact resistance, etc.) of WAA resin coating, it is not necessary to achieve the successful preparation of WAA/ nano-SiO2 nanocomposites. According to the relevant references, simple mechanical blends can also improve physical properties. Therefore, it is meaningful for us to simply adopt the physical blending method to modify nano-SiO2 to WAA resin.

Point 3: After drying (curing) polysiloxane network is expected. No evidence provided (TEM of coating film)

Response 3: We are appreciative of the reviewer’s suggestion, which will guide us in the right direction for our next research. In our future papers, we will focus on the drying (curing) polysiloxane network of WAA coating. It would have been interesting to explore this aspect. However, this manuscript mainly focuses on the untreated nano-SiO2 modified WAA resin. I hope you can continue to pay attention to my research. Special thanks to you for your valuable contribution comments.

Reviewer 3 Report

The manuscript describes the enhancement in the properties of waterborne acrylic acid resin coating by the addition of nano-silica. It concludes that the best results are obtained when the content of nano-SiO2 in the WAA coating is 1 wt%. The scientific impact of this work is low and there are some contradictions. It could meet publication standards after a thorough revision  

English grammar should be also rechecked to eliminate some errors.

1.                   The authors should add the word resin to the title (i.e. Waterborne Acrylic Acid Resin Coating).

2.                   Page 2 Lines 54-56: Please explain how the modification of WAA coatings by nano-SiO2 can effectively improve VOC.

3.                   Page 2 Line 66: If the diameter of the SiO2 nanoparticles is 30μm, they are microparticles and not nanoparticles. On the other hand, they appear as nanoparticles at the SEM images.

4.                   Page 3 Lines 132-134 “Furthermore, the peaks at 1730, 1450, and 1400 cm-1 were attributed to the absorption peaks of C=O, C-O, and -COO- in the carboxyl group of the 133 WAA coating, respectively [38].” A more detailed description of these absorption peaks is needed (especially for -COO-).

5.                   Page 2 Lines 136-139. “When SiO2 nanoparticles were added to the WAA coatings, no new FT-IR characteristic peaks were seen in the nano-SiO2/WAA coatings, indicating that no chemical reaction occurred to create the new functional group”. This comment is irrelevant. SiO2 is inert and could not react with acrylic acid. On the other hand, silica peaks do not appear in the IR spectrum. This is perhaps due to their low concentration.

6.                   Figure 1 needs a legend.

7.                   In the introduction (Page 2 lines 50-52) the authors claim that a) WAA coatings have performance defects such as low gloss. b) the modification of WAA coatings by nano-SiO2 can effectively improve the relevant physicochemical properties (such as gloss). This is not supported by the results depicted in Figure 3 where the addition of silica lowers glossiness yet the authors do not comment on this.

Author Response

Response to Reviewer 3 Comments

Thank you for giving us the opportunity to submit a revised draft of the manuscript "Nano-SiO2 Modified Waterborne Acrylic Acid Coating for Wood Wallboard" (ID: coatings-1917973) for publication in Coatings. We appreciate the time and effort that you and the reviewers dedicated to providing feedback on our manuscript and are grateful for the insightful comments on and valuable improvements to our paper.

We have incorporated most of the suggestions made by the reviewers and have made revisions which are marked in red in the manuscript. We have provided a point-by-point response to the reviewers' comments below in red color. The main corrections in the paper and the responses to the reviewer’s comments are as flowing:

Point 1: English grammar should be also rechecked to eliminate some errors.

Response 1: We are very sorry for our incorrect writing the language mistakes in this article. We have rechecked carefully the language mistakes in the article and polish it well.

Point 2: The authors should add the word resin to the title (i.e. Waterborne Acrylic Acid Resin Coating).

Response 2: We have added the word resin to the title like "Nano-SiO2 Modified Waterborne Acrylic Acid Resin Coating for Wood Wallboard".

Point 3: Page 2 Lines 54-56: Please explain how the modification of WAA coatings by nano-SiO2 can effectively improve VOC.

Response 3: Thank you for your comment. It’s true that nano-SiO2 can not effectively improve VOC, so we have deleted the word “VOC” according to relevant references.

Point 4: Page 2 Line 66: If the diameter of the SiO2 nanoparticles is 30μm, they are microparticles and not nanoparticles. On the other hand, they appear as nanoparticles at the SEM images.

Response 4: Thank you for pointing this out, and we have changed “30 μm” into “30 nm”. We feel sorry for our incorrect writing about the size unit of nano-SiO2.

Point 5: Page 3 Lines 132-134 “Furthermore, the peaks at 1730, 1450, and 1400 cm-1 were attributed to the absorption peaks of C=O, C-O, and -COO- in the carboxyl group of the 133 WAA coating, respectively [38].” A more detailed description of these absorption peaks is needed (especially for -COO-).

Response 5: Thank you for underlining this deficiency. We agree with the reviewer that a more detailed description of these absorption peaks is needed. And we have described these absorption peaks in more detail, especially for -the COO-.

Point 6: Page 2 Lines 136-139. “When SiO2 nanoparticles were added to the WAA coatings, no new FT-IR characteristic peaks were seen in the nano-SiO2/WAA coatings, indicating that no chemical reaction occurred to create the new functional group”. This comment is irrelevant. SiO2 is inert and could not react with acrylic acid. On the other hand, silica peaks do not appear in the IR spectrum. This is perhaps due to their low concentration.

Response 6: It is really a great suggestion as the Reviewer pointed out that, and after long deliberation, we have added relevant suggestions and deleted some sentences in the article.

Point 7: Figure 1 needs a legend.

Response 7: We are grateful for the suggestion. However, we have already labeled "Before" and "After" on the line as a legend, so there is no need to label the legend again.

Point 8: In the introduction (Page 2 lines 50-52) the authors claim that a) WAA coatings have performance defects such as low gloss. b) the modification of WAA coatings by nano-SiO2 can effectively improve the relevant physicochemical properties (such as gloss). This is not supported by the results depicted in Figure 3 where the addition of silica lowers glossiness yet the authors do not comment on this.

Response 8: Thank you for your valuable suggestions. As we all know, the higher the gloss value, the more reflective the surface, and the brighter the surface. In the field of wood wallboard, a low gloss value indicates that it is not bright but satisfactory matte. Therefore, a low gloss value is a good luster effect. Therefore, the data in Figure 3 can support the comments in the Introduction section by the results.

Reviewer 4 Report

The review of a manuscript “Nano-SiO2 modified waterbone acrylic acid coating for wood wallboard” submitted to Coatings Journal.

In general I find this subject interesting, there are a lot of emerging reports on the application of nanomaterials in order to reinforce some surfaces, composites, wood etc. However, I have some suggestions which are listed below:

Line 37: I don’t understand the statement “coating technology of wood panel is still less”. Is it supposed to be a justification for the aim of the study? It is presented in such short form that I don’t really understand the purpose of this sentence. If they were less study then provide the explanation why they were? What are the differences between the wallboard and other painted wooden surfaces which were previously studied in terms of this kind of protection.

Line 39: Why this needs an urgent research? My suggestion in general would be that authors should not just add the citations but summarize them, make a conclusion based on them. I can not see the explanation why this needs an urgent research based on that sentence (especially since they have been studied for many years now).

Line 42: What do you mean “traditional wood coatings” ?

Line 50: What do you mean “WAA coating has a lower economy”?

Line 57: How is that different than already published studies?

Introduction: At this point after reading the introduction I can not see any step forward in this subject. Authors should focus more on the novelty of this work. In my opinion the best way to show the new aspects which your study includes is to present the observations from other work. Despite the fact that nano-sio2 is widely studied as an additive for coatings I can not see any results in the introduction. Therefore, it should be edited to focus more on the findings not on the general, well-known information. There are some problems with nanoparticles which are well-known for now, such as agglomerates mentioned by authors in the further part of the paper, at this point in my opinion this study should provide the answers for this issues to add some knowledge.

Line 66 and 70: More details on the wallboard (photo e.g.) and nanoparticles should be presented.

Line 74-80: The provider of the equipment should be mentioned.

Line 77: What was the temperature during stirring, how fast it was stirred?

Table 1: Caption should be reconsidered. I can not see the reason for adding this numbers of samples if you are not using them as a labels for variants?

Line 87: Provide some detailed description of “crossed method”?

Line 87: How the spread rate was controlled? The error range seems to be major.

Line 89: How the samples were placed in the chamber?

Line 96: How the samples were mixed with KBr before grounding?

Line 94: What is the characteristic peak which you expected?

Line 99: Were samples spread with gold?

Line 125: How exactly could nanoparticles “destroy” the structure of WAA? Did you see some examples in literature?

In general, there is no comparison with existing literature data in case of FTIR results.

Figure 1: Caption is confusing. The labelling of samples as before and after needs to be explained. Which formulation was exactly use? There were many of them containing nano-SiO2. It should be provided in the methodology.

Line 146: What is the reason for poor dispersion in case of reference variant?

Line 155: These observed particles were agglomerates or other additives mentioned before?

Line 162: Can you give any ideas on how to prevent the formation of agglomerates? It should actually be expected during the design of your research based on previous findings. Having the access to this knowledge you should take this into consideration earlier and present some new ideas e.g. modification of nanoparticles, new ways of homogenization. Some of them are studied for years now.

Figure 2: Why there are different magnification in every picture? It is hard to compare them. The formulation which was photographed should be provided in methodology.

Line 173, 197-198, 210, 225: it is confusing way to summarize the observation.

Figure 3: Caption should be rewritten.

Line 178-179: “The abrasion… [40]” Results and Discussion is not the place for this sentence.

Line 190: If the issue was inappropriate stirring it should be repeated and investigated again during the experiments. Your results should be obtained by limiting this factors. Or maybe it should be investigated with the SEM method.

Line 213: What is the reason for not seeing the effect of agglomerates in case of hardness results? What is the reason for improvement in hardness in general?

Figure 5: There is no unit, the scale includes -0.5.

Table 2: The summary should contain the best variant which authors recommend.

Line 257: What new ideas are provided by this study? The necessity to eliminate the agglomerates and the effect of hardness or gloss is not that new idea.

Line 258-259. The last sentence is not a conclusion supported by this study.

The major problem which should not be a case in this kind of high-profile journals anymore is the lack of statistical analysis. It is a standard now and making conclusions based on the results supported just by a standard deviation is not welcome anymore. Authors should consider providing the homogenous groups etc. which will help to evaluate the significance of the differences etc. Especially in case of materials such as wooden surface, the difficulties in obtaining the homogeneous dispersions, major error in spread rate.

The linguistic corrections should be considered by the Journal.

I would like to thank authors for their effort and I hope that my suggestions will contribute to the improvement in this paper.  

Author Response

Response to Reviewer 4 Comments

On behalf of my co-authors, we thank you very much for giving us an opportunity to revise our manuscript. We appreciate you very much for their positive and constructive comments and suggestions on our manuscript entitled “Nano-SiO2 Modified Waterborne Acrylic Acid Coating for Wood Wallboard”. (ID: coatings-1917973).

We have studied your comments carefully and have made revision which marked in red in the paper. Those comments are all valuable and very helpful for revising and improving our paper, as well as the important guiding significance to our researches. The main corrections in the paper and the responds to the reviewer’s comments are as flowing:

Point 1: Line 37: I don’t understand the statement “coating technology of wood panel is still less”. Is it supposed to be a justification for the aim of the study? It is presented in such short form that I don’t really understand the purpose of this sentence. If they were less study then provide the explanation why they were? What are the differences between the wallboard and other painted wooden surfaces which were previously studied in terms of this kind of protection.

Response 1: Thank you for your valuable suggestions. And we have revised the statement “coating technology of wood panel is still less”.

Point 2: Line 39: Why this needs an urgent research? My suggestion in general would be that authors should not just add the citations but summarize them, make a conclusion based on them. I can not see the explanation why this needs an urgent research based on that sentence (especially since they have been studied for many years now).

Response 2: We are grateful for the suggestion. We have appropriately supplemented the urgency of the study in the article, and more specific instructions are in the next paragraph.

Point 3: Line 42: What do you mean “traditional wood coatings”?

Response 3: As Reviewer suggested, we have already made additional explanations as shown in the manuscript.

Point 4: Line 50: What do you mean “WAA coating has a lower economy”?

Response 4: Thank you for your comment. Because the price of WAA coating is lower than that of other waterborne coatings, we say that its economy is lower.

Point 5: Line 57: How is that different than already published studies?

Response 5: As for the referee’s concern, we believe that the different than already published studies can be easily found in the references. And we also made some supplements in the article.

Point 6: Introduction: At this point after reading the introduction I can not see any step forward in this subject. Authors should focus more on the novelty of this work. In my opinion the best way to show the new aspects which your study includes is to present the observations from other work. Despite the fact that nano-sio2 is widely studied as an additive for coatings I can not see any results in the introduction. Therefore, it should be edited to focus more on the findings not on the general, well-known information. There are some problems with nanoparticles which are well-known for now, such as agglomerates mentioned by authors in the further part of the paper, at this point in my opinion this study should provide the answers for this issues to add some knowledge.

Response 6: We are appreciative of the reviewer’s suggestion. Indeed, it will be more profound if we focus more on the novelty of this work.  And we have tried our best to revised this part. Thanks again for your valuable comment.

Point 7: Line 66 and 70: More details on the wallboard (photo e.g.) and nanoparticles should be presented.

Response 7: Thank you very much to point out the more details on the wallboard and nanoparticles. The wallboard and nanoparticles used in our experiments are common materials on the market, and you can easily find more detailed information on the Internet. Therefore, we do not think it is necessary to present in the article.

Point 8: Line 74-80: The provider of the equipment should be mentioned.

Response 8: Considering the Reviewer’s suggestion, we have mentioned the provider of the equipment in the manuscript.

Point 9: Line 77: What was the temperature during stirring, how fast it was stirred?

Response 9: Thanks for your valuable comment and we have made some revised. It was stirred at 500 r/min for 2 h at room temperature to make the dispersion uniform. And we have added this information to the manuscript.

Point 10: Table 1: Caption should be reconsidered. I can not see the reason for adding this numbers of samples if you are not using them as a labels for variants?

Response 10: As suggested by the reviewer, we have revised the caption of Table 1.

Point 11: Line 87: Provide some detailed description of “crossed method”?

Response 11: As requested by the reviewer, we have provided some detailed description of “crossed method”.

Point 12: Line 87: How the spread rate was controlled? The error range seems to be major.

Response 12: We are grateful for the suggestion, actually we can control the spread rate by adjusting the pressure on the spray gun. Because the paint output of the paint gun is greatly influenced by subjective factors, it is normal for the error range to be major.

Point 13: Line 89: How the samples were placed in the chamber?

Response 13: We are grateful for the suggestion and we placed the samples in the chamber by forceps. And we have added this information in the manuscript.

Point 14: Line 96: How the samples were mixed with KBr before grounding?

Response 14: As for the Reviewer’s concern, the samples were mixed with KBr in a mortar, and we have added this information in the sentence.

Point 15: Line 94: What is the characteristic peak which you expected?

Response 15: As for the Reviewer’s concern, the characteristic peak which I expect can be find in the 3. Results and Discussion section.

Point 16: Line 99: Were samples spread with gold?

Response 16: Yes. Spraying gold is a conventional method for SEM, so we do not explain further in the article, and everyone knows this process by default. We can add to the manuscript if you need.

Point 17: Line 125: How exactly could nanoparticles “destroy” the structure of WAA? Did you see some examples in literature?

Response 17: We are grateful for the suggestion. However, the original words in the manuscript is “which indicates that the modification of nano-SiO2 did not destroy the basic structure of the WAA coatings by chemical reaction.” What I mean in the manuscript is that the structure of the WAA is not destroy. So, I don't understand why the reviewer said the nanoparticles “destroy” the structure of WAA.

Point 18: In general, there is no comparison with existing literature data in case of FTIR results.

Response 18: Thank you for pointing out the problem, we have added some comparison with existing literature data in case of FTIR results.

Point 19: Figure 1: Caption is confusing. The labelling of samples as before and after needs to be explained. Which formulation was exactly use? There were many of them containing nano-SiO2. It should be provided in the methodology.

Response 19: Thank you for your careful work. We have revised the caption of Figure 1. Meanwhile, we also supplement the ormulation we use and provided in the methodology.

Point 20: Line 146: What is the reason for poor dispersion in case of reference variant?

Response 20: Thank you for pointing this out. The main reason for the deterioration of dispersion performance is that the agglomeration effect of nanoparticles is aggravated by the addition of too much nano-SiO2.

Point 21: Line 155: These observed particles were agglomerates or other additives mentioned before?

Response 21: We are grateful for the valuable suggestion. According to the relevant references, these observed particles were agglomerates rather than other additives.

Point 22: Line 162: Can you give any ideas on how to prevent the formation of agglomerates? It should actually be expected during the design of your research based on previous findings. Having the access to this knowledge you should take this into consideration earlier and present some new ideas e.g. modification of nanoparticles, new ways of homogenization. Some of them are studied for years now.

Response 22: It is really true as Reviewer suggested that to prevent the formation of agglomerates is necessary. Thank you for your valuable suggestions, which will guide us in the right direction for our next research. In our future papers, we will focus on the modification of nanoparticles to prevent the formation of agglomerates. It would have been interesting to explore this aspect. However, this manuscript mainly focuses on the untreated nano-SiO2 modified WAA resin. I hope you can continue to pay attention to my research. Special thanks to you for your valuable contribution comments.

Point 23: Figure 2: Why there are different magnification in every picture? It is hard to compare them. The formulation which was photographed should be provided in methodology.

Response 23: We are very sorry for the mistake we have made in Figure 2. We'll pay attention to this issue in our future research. Moreover, we have provided the formulation which was photographed in methodology.

Point 24: Line 173, 197-198, 210, 225: it is confusing way to summarize the observation.

Response 24: We are very sorry for our inappropriate writing about summarize of the observation of this manuscript. We have re-written the part according to the Reviewer’s suggestion.

Point 25: Figure 3: Caption should be rewritten.

Response 25: Thank you very much to point out the caption of Figure 3. We have rewritten the caption.

Point 26: Line 178-179: “The abrasion… [40]” Results and Discussion is not the place for this sentence.

Response 26: Thank you for pointing this out. We are very sorry for our incorrect writing about the Results and Discussion. And we have change the place for this sentence.

Point 27: Line 190: If the issue was inappropriate stirring it should be repeated and investigated again during the experiments. Your results should be obtained by limiting this factors. Or maybe it should be investigated with the SEM method.

Response 27: Thank you for pointing this out. We have rewritten this part.

Point 28: Line 213: What is the reason for not seeing the effect of agglomerates in case of hardness results? What is the reason for improvement in hardness in general?

Response 28: We thank you for your careful read and thought on our manuscript. As we know, nano-SiO2 is a nano-material with a crystal effect, so adding it to WAA resin can improve the hardness of the coating. So, we have added this sentence into Lin 213.

Point 29: Figure 5: There is no unit, the scale includes -0.5.

Response 29: We gratefully appreciate your valuable comment. We have added the unit and deleted the scale -0.5.

Point 30: Table 2: The summary should contain the best variant which authors recommend.

Response 30: Thank you for this suggestion. We have the best variant which we recommend in Table 2.

Point 31: Line 257: What new ideas are provided by this study? The necessity to eliminate the agglomerates and the effect of hardness or gloss is not that new idea.

Response 31: We feel sorry for the inconvenience brought to the reviewer, because of the wrong word “idea”. And we have revised this sentence in this manuscript.

Point 32: Line 258-259. The last sentence is not a conclusion supported by this study.

Response 32: Thank you for pointing this out. Considering the Reviewer's suggestion, we have removed the last sentence.

Point 33: The major problem which should not be a case in this kind of high-profile journal anymore is the lack of statistical analysis. It is a standard now and making conclusions based on the results supported just by a standard deviation is not welcome anymore. Authors should consider providing the homogenous groups etc. which will help to evaluate the significance of the differences etc. Especially in the case of materials such as wooden surfaces, the difficulties in obtaining homogeneous dispersions, and major errors in spread rate.

Response 33: We appreciate it very much for this good suggestion. It is true that statistical analysis is a standard for evaluating the significance of the differences. In our future research, we will pay much attention to statistical analysis. We hope you can continue to focus on our future research. Thanks again for your valuable comment.

Point 34: The linguistic corrections should be considered by the Journal.

Response 34: We are very sorry for the grammar mistakes in this manuscript and the inconvenience they caused in your reading. In order to meet the reviewer's requirements, we have our manuscript careful editing by someone with expertise in technical English editing paying particular attention to English grammar, spelling, and sentence structure so that the goals and results of the study are clear to the reader.

Round 2

Reviewer 1 Report

The authors have upgraded the requested parts, nothing to add.

Author Response

We appreciate you very much for your positive and constructive comments and suggestions on our manuscript.

Thanks again.

Reviewer 2 Report

Point 1. In the material section (version v1), authors provided the aver. size of 30 micron. But then it is changed to 30 nm in version v2.  That is sceptical information that authors didn't answer to my previous comment but author said that it was commercial product. Coatings is a research journal which aims to present scientific knowledge.

Point 2. Section 2.1 ( Preparation of nano-SiO2 modified WAA coating). The modification means changes by chemical or physical interactions in views of scientific community. The goal of this part is to achieve "uniform dispersion (line 80). For scientific interest, we are keen to know how authors proved that happened successfully not just the results of testings (commercial interest). Noted that section 2.1 titled "Preparation of nano-SiO2 modified WAA coating" but Table 1 titled "WAA and nano-SiO2 mixing ratio". It is contradictory between "modification" and "mixing". So for scientific research we need instrumental results to prove the hypothesis. 

Author Response

On behalf of my co-authors, we thank you very much for giving us an opportunity to revise our manuscript. We appreciate you very much for positive and constructive comments and suggestions on our manuscript entitled “Nano-SiO2 Modified Waterborne Acrylic Acid Coating for Wood Wallboard”. (ID: coatings-1917973).

We have studied your comments carefully and have made revision which marked in red in the paper. Those comments are all valuable and very helpful for revising and improving our paper, as well as the important guiding significance to our researches. The main corrections in the paper and the responds to the reviewer’s comments are as flowing:

Point 1: In the material section (version v1), authors provided the aver. size of 30 micron. But then it is changed to 30 nm in version v2.  That is sceptical information that authors didn't answer to my previous comment but author said that it was commercial product. Coatings is a research journal which aims to present scientific knowledge.

Response 1: Thank you again for your valuable suggestions. However, we think it isn’t necessary to do further characterization on nano-SiO2. We can easily find its detailed information from its manufacturer, we don't need to waste more time to test and characterize it.

Point 2: Section 2.1 ( Preparation of nano-SiO2 modified WAA coating). The modification means changes by chemical or physical interactions in views of scientific community. The goal of this part is to achieve "uniform dispersion (line 80). For scientific interest, we are keen to know how authors proved that happened successfully not just the results of testings (commercial interest). Noted that section 2.1 titled "Preparation of nano-SiO2 modified WAA coating" but Table 1 titled "WAA and nano-SiO2 mixing ratio". It is contradictory between "modification" and "mixing". So for scientific research we need instrumental results to prove the hypothesis.

Response 2: We thanks for your careful read and thoughtful on our manuscript. We are grateful for the suggestion. And as for the Section 2.1, we have used FT-IR and SEM test methods proved the modification happened successfully. Meanwhil, We have revised the Table 1 titled "WAA and nano-SiO2 mixing ratio" into "The proportion of nano-SiO2 in WAA resin in different samples".

Reviewer 3 Report

The authors politely addressed many of the problematic issues of the manuscript. 1) Please change Resign with Resin 2) Furthermore, since a low gloss value is a good luster effect it should not be presented as a "performance defect" (Page 2 Line 57).

Author Response

We are appreciative of the reviewer’s suggestion, we have changed Resign with Resin. Meanwhile, we have deleted the word "low gloss ".

Thanks again.